# A Two-Step Approach Using the National Health Institutes of Health Stroke Scale Assessed by Paramedics to Enhance Prehospital Stroke Detection: A Case Report and Concept Proposal

**DOI:** 10.3390/jcm13175233

**Published:** 2024-09-04

**Authors:** Loric Stuby, Mélanie Suppan, Thibaut Desmettre, Emmanuel Carrera, Matthieu Genoud, Laurent Suppan

**Affiliations:** 1Genève TEAM Ambulances, Emergency Medical Services, 1201 Geneva, Switzerland; 2Division of Anesthesiology, Department of Anesthesiology, Clinical Pharmacology, Intensive Care and Emergency Medicine, Geneva University Hospitals and Faculty of Medicine, 1205 Geneva, Switzerland; melanie.suppan@hug.ch; 3Division of Emergency Medicine, Department of Anesthesiology, Clinical Pharmacology, Intensive Care and Emergency Medicine, Geneva University Hospitals and Faculty of Medicine, 1205 Geneva, Switzerland; thibaut.desmettre@hug.ch (T.D.); laurent.suppan@hug.ch (L.S.); 4Stroke Center, Department of Neurology, Geneva University Hospitals and Faculty of Medicine, 1205 Geneva, Switzerland; emmanuel.carrera@hug.ch

**Keywords:** case report, NIHSS, paramedic, posterior circulation stroke, hemianopia, prehospital, strategy, emergency medical services

## Abstract

**Background:** Prehospital detection and triage of stroke patients mostly rely on the use of large vessel occlusion prediction scales to decrease onsite time. These quick but simplified scores, though useful, prevent prehospital providers from detecting posterior strokes and isolated symptoms such as limb ataxia or hemianopia. **Case report:** In the present case, an ambulance was dispatched to a 46-year-old man known for ophthalmic migraines and high blood pressure, who presented isolated visual symptoms different from those associated with his usual migraine attacks. Although the assessment advocated by the prehospital guideline was negative for stroke, the paramedic who assessed the patient was one of the few trained in the National Institutes of Health Stroke Scale assessment. Based on this assessment, the paramedic activated the fast-track stroke alarm and an ischemic stroke in the right temporal lobe was finally confirmed by magnetic resonance imaging. **Discussion and conclusions:** Current prehospital practice enables paramedics to detect anterior strokes but often limits the detection of posterior events or more subtle symptoms. Failure to identify such strokes delay or even forestall the initiation of thrombolytic therapy, thereby worsening patient outcomes. We therefore advocate a two-step prehospital approach: first, to avoid unnecessary delays, the prehospital stroke assessment should be carried out using a fast large vessel occlusion prediction scale; then, if this assessment is negative but potential stroke symptoms are present, a full National Institutes of Health Stroke Scale assessment could be performed to detect neurological deficits overlooked by the fast stroke scale.

## 1. Introduction

Strokes are one of the few time-sensitive emergencies. Emergency medical services (EMSs) are an important link in the stroke chain of recovery as prehospital actions are associated with better outcomes [1]. To improve functional outcomes, fast and accurate prehospital stroke detection is therefore essential [2]. While some presentations are clinically characteristic, others can be challenging, particularly as resources are limited in the prehospital setting. Although guidelines, tools, and systems designed to detect stroke continue to evolve, only a few large vessel occlusion (LVO) prediction scales have been studied in the prehospital setting [3,4]. These scores present slight differences [5,6,7,8,9,10,11,12], but none have proven to be clearly superior to others in the prehospital field [13,14].

Stroke management focuses on rapid reperfusion with intravenous thrombolysis and endovascular thrombectomy, which are both effective to reduce disability but are time-critical. Accordingly, improving the system of care to reduce treatment delays is key to maximizing the benefits of reperfusion therapies. Intravenous thrombolysis should be administered within 4.5 hours after the onset of stroke (up to 9 h in certain cases), while endovascular thrombectomy should be performed within 6 h (up to 24 h in certain cases) [15]. Some scoring systems including more cortical signs (e.g., gaze, visual fields, aphasia, and neglect), such as the National Institutes of Health Stroke Scale (NIHSS), showed better accuracy to predict stroke. The evaluation of the NIHSS symptom profile may help to stratify patients’ risk of anterior LVO and to identify individuals who deserve rapid transfer to comprehensive stroke centers [16]. However, the assessment of these signs could be difficult to investigate by paramedics and EMS personnel [17]. The aim of the present case is to report the use of the NIHSS by a paramedic to detect a stroke overlooked by the regular prehospital stroke detection score and to propose a concept of prehospital strategy using a two-step approach.

## 2. Case Report

We report the case of a 46-year-old man with a medical history of ophthalmic migraine attacks manifested by scotomas and headaches, and high blood pressure treated with telmisartan. The timeline is summarized in Figure 1. The patient was at work in Geneva, Switzerland, when he presented with sudden lipothymia associated with visual impairment. An ambulance was dispatched at 05:08 PM by the emergency medical call center (emergency telephone number 144) 10 min after the episode with the following alarm message: “Had a malaise // had visual disturbance on one side // fades // history of migraine with aura but different this time”. The emergency medical call center dispatched an ambulance staffed by a paramedic and a third-year student paramedic without using the priority signals, for an on-site assessment within 20 min. The ambulance crew arrived on the scene 17 min after the onset of symptoms.

The patient described the sudden visual loss as the inability to see the person standing on his left side. No major focal neurological deficit was identified according to the EMS stroke detection score, i.e., a gaze–face–arm–speech–time (G-FAST) score of 0. No headache was reported, and the crew asked the patient to walk to the ambulance to carry out a more detailed assessment and determine whether transport to a health care setting, or even to a neurovascular facility, would be necessary.

The following vital parameters were measured: pulse rate was 60 beats per minute, blood pressure 137/87 mmHg, peripheral oxygen saturation 97%, blood glucose 4.7 mmol/L (84.6 mg/dL), and body temperature 36° Celsius. The patient reported no complaints other than the discomfort due to the visual loss. The paramedic, who had been trained in the NIHSS assessment, found a score of 1 by identifying a partial homonymous hemianopia (left lower quadranopsia). Although the patient suffered from visual field amputations due to his previous migraines, he clearly stated that the symptoms were different from his usual migraine attacks. On this basis, the paramedic activated the fast-track stroke alarm; under normal circumstances, the prehospital alarm is triggered either on the basis of the G-FAST [7,8,18] or by the presence of dizziness associated with the inability to sit or stand since less than 8 h (or at wake-up). This prehospital alarm alerts a comprehensive team, consisting of a neurologist, an emergency physician, and an emergency nurse practitioner, who welcome the patient when the ambulance arrives. The aim is to reduce the door-to-needle (DTN) and/or door-to-puncture times thanks to direct admission to the computed tomography scanner (CT scan) should the suspicion of a stroke be confirmed by the team [19]. In that case, the ambulance left the site 15 min after contact with the patient, and reached the university hospital (stroke center) 54 min after the onset of symptoms. It is worth mentioning that the local procedure prevents direct communication between the hospital (in this case, the on-call neurologist) and the paramedic in charge.

The neurologist who assessed the patient at the door found an NIHSS score of 2, concluding to a left homonymous lateral hemianopia. The stroke fast-track was therefore confirmed, and the patient was directly brought to the scanner from the prehospital stretcher 12 min after hospital admission. The CT scan showed no ischemic lesion, no perfusion asymmetry, and no vascular occlusion or stenosis. On clinical reassessment, the deficit had completely regressed, and intravenous thrombolysis was therefore unnecessary.

A more detailed history was thereafter obtained. The patient indicated that the migraine episodes had become more frequent over the last 18 months, at a rate of once every four months, lasting between 30 min and 72 h at most. Auras occurred without any associated headache. Non-steroidal anti-inflammatory painkillers were effective to treat the symptoms. Following complete resolution of the symptoms, the patient returned home with a prescription of outpatient brain magnetic resonance imaging (MRI).

Three days later, the patient underwent an outpatient MRI scan which revealed an acute punctiform ischemic lesion in the right hippocampus. Following these findings on the MRI, the patient was admitted to the stroke unit for further investigations and etiological work-up. Secondary prevention with double antiplatelet therapy and statin was initiated. The patient was subsequently referred to a neuro-ophthalmologist who described two additional ipsilateral punctiform cortical lesions in the occipital lobe. These lesions are all suitable with an embolic phenomenon through the right posterior cerebral artery.

## 3. Discussion and Concept Proposal

Most isolated homonymous hemianopias are caused by ischemic lesions in the occipital lobe that generally do not produce other neurologic manifestations [20,21], and are thus not detected by simple scores such as the Cincinnati Prehospital Stroke Scale (CPSS) or the G-FAST. In the setting presented, the G-FAST is the only score integrated to the paramedics’ guidelines, and a prehospital stroke alarm would certainly not have been activated if the paramedic had not co-created an e-learning course on the application of this score [22], and studied the impact of this teaching method on different populations (paramedics, medical students, and neurology department staff) [23,24,25].

Simple scores, such as the CPSS or the G-FAST, have high sensitivity to quickly detect major anterior LVO symptoms [6,7], but their specificity is limited [13]. As commented by Purington and colleagues, EMS providers are not trained to focus on outliers [26]. Thus, while prehospital stroke scales detect anterior LVO with acceptable-to-good accuracy [8], they can miss up to 30% of acute strokes [27]. Their performance is much poorer when considering posterior circulation strokes [28,29,30,31]. Such strokes are more often initially misdiagnosed than anterior strokes [32,33], and are associated with prolonged DTN time [34] and worse outcomes [35]. Additionally, thrombolysis rates are 50% lower in posterior circulation strokes [36], and relatively few ischemic stroke patients with isolated homonymous hemianopia receive thrombolysis, even though it has been shown to be both safe and effective [37,38].

Some authors have proposed using shorter and quicker NIHSS versions in the prehospital setting [39,40,41,42], but performing the original full 15-item version is still necessary to assess stroke severity, as it provides more discriminative information and is more responsive to changes in the neurologic status than the shortened versions [43]. Teaching the NIHSS to paramedics could be carried out rather easily and efficiently using an e-learning module [23]. Paramedics could then use the NIHSS as an accurate and time-efficient prehospital stroke severity quantification tool, enabling the evaluation of stroke evolution since the prehospital stage, and providing a common language for stroke assessment between paramedics and stroke physicians [44]. The need to use the same scale in emergency settings, both prehospital and in-hospital, has already been advocated to allow for a standardized approach, improve communication, and reduce potential misunderstandings [14]. Although the use of the NIHSS in the prehospital environment results in a slight increase in on-site time (2 to 5 min), it does not increase DTN time [44,45]. A very recent study by Claudie et al. promoted the FAST4D score, which incorporates additional items such as the acute onset of diplopic images, deficit in the field of vision, dizziness/vertigo, and dysmetria/ataxia. This score demonstrated an increase in sensitivity compared to the traditional FAST score (15% increase in stroke detection), and reduced false negatives by 65%. Given these promising results, the FAST4D score deserves further attention and should be disseminated widely, as it may also serve as a potential alternative to the NIHSS [46]. Another avenue deserving further attention is the impact of telemedicine in these cases, with a hospital-based neurologist performing the patient’s NIHSS assessment remotely using dedicated telemedicine equipment [47].

As time until arrival to hospital is longer and thrombolysis is delayed in patients with posterior circulation stroke or in cases of more subtle symptoms, there is a clear need to increase awareness regarding these delays, while avoiding over-triage as much as possible. As noted by Purington and colleagues, maximal stroke activation for every case of isolated vertigo would create many false positives, thus negatively impacting the timeframe for anterior LVO strokes [26]. One of the aims of advanced prehospital stroke care is to reduce the subsequent workload in emergency departments [48]. Improving outcomes in such patients will certainly require the use of scales focusing on more subtle clinical symptoms already at the prehospital stage. This could indeed increase the proportion of patients with posterior circulation stroke receiving thrombolytic treatment [49].

Another point is that, in our opinion, direct communication between the hospital (i.e., the neurologist on call) and the paramedic in charge, as this is the case in other systems even in Switzerland, could be beneficial. This could both save time and limit over-triage. Time would be saved because the neurologist would already be informed (at least roughly) of the clinical presentation. In addition, it would enable the on-call neurologist to withhold unnecessary fast-tracks even before hospital admission, therefore saving resources which would otherwise be mobilized unnecessarily. Additionally, discrepancies between EMS policies or protocols have already been acknowledged, highlighting the need for standardization to improve overall efficiency and consistency [50].

We subsidiarily think that a first-line MRI would probably have identified this multifocal ischemic stroke earlier, allowing faster admission to the neurovascular unit and avoiding diagnostic uncertainty during the three days of ambulatory wandering. We will not emphasize the diagnostic value and widespread adoption of this modality which is widely described in the literature [51,52,53,54,55,56,57].

Finally, emergency department staff rely on EMS stroke assessments in decision-making procedures. Failure to notify the hospital about stroke presentation could lead to extended delays (e.g., patient arriving via EMSs without pre-notification suffering from longer delays to in-hospital stroke-team activation compared to those arriving by private vehicles [58]). This over-reliance of emergency department personnel on EMSs for stroke recognition and activation of in-hospital resources highlights the need for more accurate prehospital triage.

A two-step approach could thus be considered to improve stroke detection without delaying LVO treatment: The first step should still consist of the application of a simple stroke scale designed to detect anterior strokes. If this scale is negative but neurological symptoms are present, a full NIHSS assessment should then be performed to enable the detection of more subtle signs (hemianopia, limb ataxia, or neglect). In this respect, further prehospital research is required to improve the diagnostic utility of the NIHSS applied by paramedics.

## 4. Limitations

The main limitation of this case report and concept proposal is that it focuses on a single case. Thus, generalizability cannot be taken for granted and large-scale studies will need to assess whether our concept can be implemented. In addition, the absence of a control group and the lack of statistical analysis limit the possibility of drawing definitive conclusions or establishing causality. Furthermore, a risk of selection and publication bias cannot be excluded. Consequently, this report should only be considered as a basis for further research, not as conclusive proof that the proposed concept should indeed be implemented. The main strength of this paper lies in the detailed presentation of a rather unique case, which perfectly reflects our proposed concept of NIHSS integration in the prehospital setting.

Although the NIHSS may help detect more strokes, it appears to be an insufficient tool for screening posterior circulation lesions. For patients presenting with acute isolated dizziness, clinicians are thus faced with the need to adapt their clinical scores, including specific sequential maneuvers (head impulse, nystagmus, test of skew, HINTS, or the same with the addition of a simple hearing test, i.e., finger rub test, HINTS+) or expanding classical scales with additional tests (i.e., examination of truncal ataxia or gait stability in the posterior NIHSS [59]). This point is of crucial importance because dizziness is the symptom most tightly linked to missed ischemic strokes [60] and to diagnostic delays [61]. In patients presenting with acute vestibular syndrome (AVS), a HINTS/+ examination should be considered if a spontaneous nystagmus is seen during the examination; whilst HINTS performs better than MRI in detecting stroke [62], it nevertheless requires both time and clinical expertise. In the absence of spontaneous nystagmus, the evaluation of gait unsteadiness or truncal ataxia has a positive predictive value for stroke, depending on severity grade [63,64,65]. The inability to sit is currently used in our prehospital setting as a surrogate for truncal ataxia [66]. While major deficits are usually clinically obvious and justify reperfusion treatments, there is considerable uncertainty regarding AVS with no or mild deficits [67]. Moreover, there is currently no guidance as to whether a patient meeting the “central” HINTS criteria should receive thrombolysis even when presenting within the therapeutic window [68]. Thereupon, it would probably make more sense, among patients with isolated AVS without any other deficit (i.e., negative G-FAST and NIHSS, and without truncal ataxia, which are all independent criteria for triggering the alarm), to only look for spontaneous nystagmus, and if present, to trigger the prehospital fast-track alarm to allow for a fast HINTS+ assessment at the hospital’s door.

## 5. Conclusions

Implementing the NIHSS in the prehospital setting in a two-step algorithm could enhance stroke detection by paramedics, thus improving the triage of patients by detecting neurological symptoms missed by usual scores.

## 6. Patient Perspective

The patient’s perspective was directly written in English by the patient, without any modification by the authors:


*“The above constitutes an accurate narrative of my patient journey. Herewith however a few additional details:*

*- I moved twice until I got into the ambulance, and both times I walked rather than being on a stretcher,*

*- When I was invited to return home, (a) I asked if I could ride my motorbike home and was told “yes” (I however decided to take public transport as I did not feel well and did not trust my eyesight fully), and (b) I was invited to do an MRI “in the coming three months.”*

*- I was told on Thursday morning of that week, by the radiologist interpreting the MRI (which I did the day before in Zurich), that I had a lesion which was congruent with the temporary loss of sight. However, it took until Friday 5 pm (and at least 10 calls from me to the hospital asking for follow up) for a fairly stressed-out neurologist to call me up and tell me I had to come to the hospital immediately.*

*What these additional details indicate to me is that I went from high-risk to no-risk status on the basis of one CT scan reading alone, which means that any error in the diagnosis has a greater potential for harm due to the lack of follow up. The MRI should have been planned for the next day, in Geneva, and should have been a follow up of the same “case” handled by the hospital, i.e., high-risk. Certain preventive measures should have been advised, e.g., In my case: (i) do not drive a personal vehicle until a full visual test is done, (ii) test your limbs, eyesight and speech (G-FAST or similar) every 4 h, (iii) do not venture outside alone for the next few days and stay within 1 hr of a hospital.*

*Indeed, I took the train to go to Zurick for the MRI on Wednesday, and on Thursday I went for a 75 min run in the woods, all alone, before I got the call from the Radiologist. If I had listened to the hospital staff, I would have taken my motorbike to ride home (15 km away, by night). I feel very lucky to have gone through that period of risk and uncertainty unscathed.”*


## Figures and Tables

**Figure 1 jcm-13-05233-f001:**
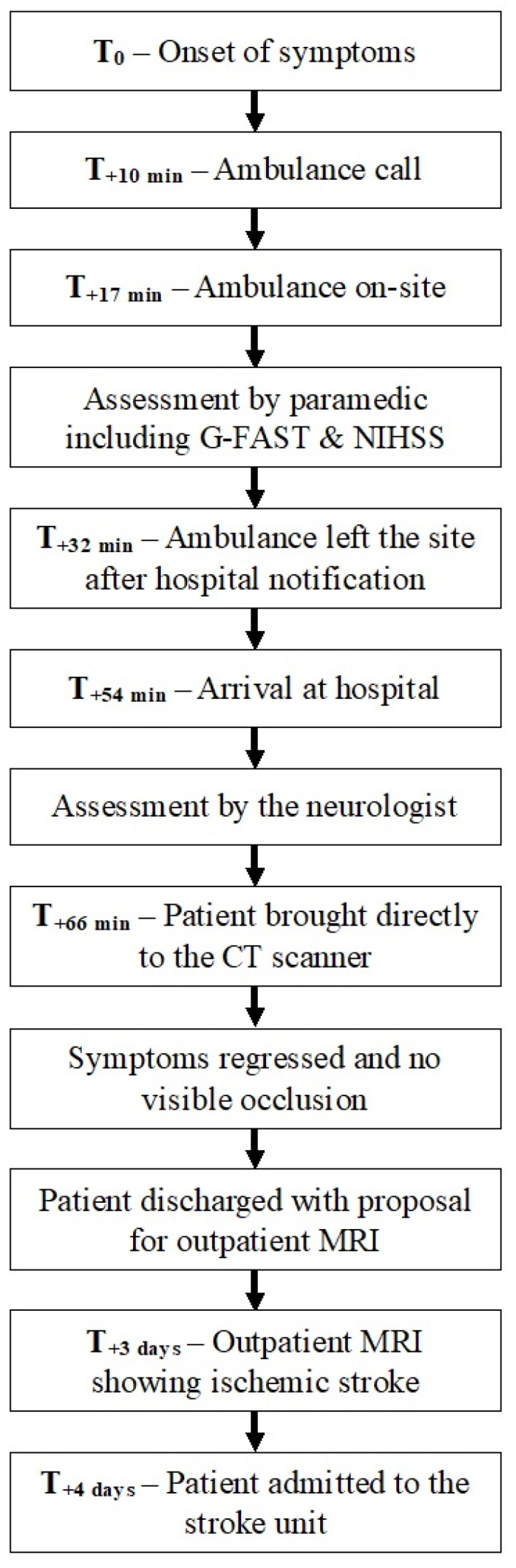
Timeline.

## Data Availability

The original contributions presented in the study are included in the article.

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
