# Peer review of "A Two-Step Approach Using the National Health Institutes of Health Stroke Scale Assessed by Paramedics to Enhance Prehospital Stroke Detection: A Case Report and Concept Proposal"

_jcm, 2024, doi:10.3390/jcm13175233_

Round 1

Reviewer 1 Report

Comments and Suggestions for Authors

Who is the comprehensive team made up of? Only from the neurologist? (this must be specified)

What other health problems did the patient have? What background medication did he have at home?

Change subchapter Detailed case description with Case Report

Why was the patient not hospitalized/kept under surveillance?

The limitations of the study are very well written.

You need to discuss stroke time targets and code stroke a little more.

Discuss also other recent articles regarding acute stroke management:

a)     doi: 10.3390/medicina58111541

b)      https://doi.org/10.3390/jpm14060596

c)     doi: 10.1186/s13049-017-0377-x

d)     https://doi.org/10.3390/jpm14010013

Reviewer 2 Report

Comments and Suggestions for Authors

Thank you for the opportunity to read an interesting manuscript. The problem of recognizing stroke in pre-hospital conditions is significant and few studies assess the effectiveness of scales in pre-hospital conditions. However, I have significant doubts about the form of presenting this topic in the article. The described situation represents one of the relatively rare cases where a very low score on the NIHSS scale also gives the possibility of suspecting stroke. A score of 1 is not an indication for urgent intervention, which the paramedic decided on. No other scale was used to verify the patient in pre-hospital conditions. It should be noted that the patient also had a negative CT result. It is well known that ischemic changes appear in CT only after several to a dozen or so hours, so this is not a surprising symptom. Secondary imaging (MRI) repeatedly confirms such situations. However, this does not result in urgent thrombolysis or thrombectomy, but conservative treatment. Therefore, I believe that the innovativeness of the described case is low.

However, I will provide a few specific comments to the authors in case of further proceedings:

1) The purpose of the study should be changed/specified.

2) The place/region and time in which the event took place should be provided. Since this is a case report, I recommend using the CARE diagram.

3) The final conclusions should be expanded.

4) Many of the literature items are older than the last 5 years. I recommend supplementing the list with the following publications:

a) Furey P, Town A, Sumera K, Webster CA. Approaches for integrating generative artificial intelligence in emergency healthcare education within higher education: a scoping review. Crit. Care Innov. 2024; 7(2): 34-54.

(the publication shows modern forms of education for paramedics, which refers to the e-learning course implemented by the paramedic in the case report)

b) Świeżewski SP, Rzońca P, Panczyk M, Leszczyński PK, Gujski M, Michalak G, Fronczak A, Gałązkowski R. Polish Helicopter Emergency Medical Service (HEMS) Response to Stroke: A Five-Year Retrospective Study. Med Sci Monit. 2019; 25:6547-6553.

(the publication shows the scale of HEMS interventions for stroke patients, which refers to the seriousness of the problem raised by the authors of the case report)

Reviewer 3 Report

Comments and Suggestions for Authors

I would like to thank the authors for the opportunity they have given me to read this interesting case report. I find it an interesting and well-described case. Since pre-hospital care is not the same in all countries, I find it interesting to know some aspects of the activation of the transfer to the neurovascular center useful. In some countries, the activation of this "stroke code" emergency by the pre-hospital emergency system requires a telephone call from the paramedic (or even the pre-hospital doctor) to the neurologist (or neurovascular manager of the hospital) to accept the transfer (ensuring that the case meets the agreed criteria). This call is immediately managed by the emergency coordination center and the neurologist (for immediate location). I would like the authors to comment on this aspect of their case. In their case, was there this prior communication (hospital advance notice) between the paramedic and the hospital? What is the opinion of the authors? Would this alert system, as it exists in some countries, be necessary?

Round 2

Reviewer 2 Report

Comments and Suggestions for Authors

the comments requested by me were not included.

Author Response

The reviewer's comment has been addressed in the initial response, with supporting arguments provided. No further modifications were made during this second round.